

# CHLSOC: The Chilean Soil Organic Carbon database, a multi-institutional collaborative effort

Marco Pfeiffer[1], José Padarian[2], Rodrigo Osorio[3], Nelson Bustamante[3], Guillermo Federico Olmedo[4,5], Mario Guevara[6], Felipe Aburto[7*], Monica Antilen[8,9*], Elías Araya[3*], Eduardo Arellano[10*], Maialen Barret[11*], Juan Barrera[12*], Pascal Boeckx[13*], Margarita Briceño[14*], Sally Bunning[15*], Lea Cabrol[16,17*], Manuel Casanova[1*], Pablo Cornejo[18*], Fabio Corradini[19*], Gustavo Curaqueo[20*], Sebastian Doetterl[21*], Paola Duran[18*], Mauricio Escudey[22,9*], Angelina Espinoza[23*], Samuel Francke[24*], Juan Pablo Fuentes[25*], Marcel Fuentes[26*], Gonzalo Gajardo[27*], Rafael García[7*], Audrey Gallaud[27*], Mauricio Galleguillos[28*], Andres Gomez[3*], Marcela Hidalgo[12*], Jorge Ivelic-Sáez[29*], Lwando Mashalaba[1*], Francisco Matus[18*], Maria de la Luz Mora[18*], Jorge Mora[30*], Cristina Muñoz[12*], Pablo Norambuena[31*], Carolina Olivera[32*], Carlos Ovalle[33*], Marcelo Panichini[34*], Pauchard Aníbal[7*], Jorge F. Perez-Quezada[28, 39*], Sergio Radic[35*], José Ramirez[36*], Nicolas Riveras[1*], German Ruiz[3*], Osvaldo Salazar[1*], Ivan Salgado[3*], Oscar Seguel[1*], Maria Sepúlveda[12*], Carlos Sierra[26*], Yasna Tapia[1*], Balfredo Toledo[27*], José Miguel Torrico[37*], Susana Valle[38*], Ronald Vargas[4*], Michael Wolff[34*], and Erick Zagal[12*]

*These authors contributed equally to this work
[1]Departamento de Ingenieria y Suelos, Facultad de Ciencias Agronómicas, Universidad de Chile. Santa Rosa 11315, La Pintana, Chile
[2]Faculty of Agriculture and Environment, The University of Sydney, New South Wales, Australia
[3]Servicio Agrícola y Ganadero (SAG), Ministerio de Agricultura, Av. Presidente Bulnes 140, Santiago, Chile
[4]Food and Agriculture Organization of the United Nations (FAO), Viale delle Terme di Caracalla, Rome, Italy
[5]Instituto Nacional de Tecnología Agropecuaria (INTA) Mendoza, San Martín 3853, Luján de Cuyo, Mendoza, Argentina
[6]Department of Plant and Soil Sciences. University of Delaware, Newark, DE 19716, USA
[7]Laboratorio de Investigación en Suelos, Aguas y Bosques (LISAB), Universidad de Concepción, Departamento de Silvicultura, Facultad de Ciencias Forestales, Victoria 631, Concepción, Chile
[8]Departamento de Química Inorgánica, Facultad de Química y de Farmacia, Pontificia Universidad Católica de Chile, Vicuña Mackenna 4860, Santiago, Chile
[9]Centro para el Desarrollo de la Nanociencia y Nanotecnología (CEDENNA), Av. L.B. O'Higgins 3363, Santiago, 7254758, Chile
[10]Departamento de Ecosistemas y Medio Ambiente, Facultad de Agronomía e Ingeniería Forestal, Pontificia Universidad Católica de Chile
[11]University of Toulouse, CNRS, INP, Ecolab UMR5245, Toulouse, France
[12]Departamento de Suelos y Recursos Naturales, Universidad de Concepción, Campus Chillán, Vicente Méndez 595, Chillán, Chile
[13]Isotope Bioscience Laboratory, Ghent University, Gent, Belgium
[14]Facultad de Ciencias de la Salud, Universidad Arturo Prat, Av. Arturo Prat 2120, Iquique, Chile
[15]Food and Agriculture Organization of the United Nations (FAO), Regional Office for Latin America and the Caribbean, Dag Hammarskjold, Vitacura, Chile
[16]Escuela de Ingeniería Bioquímica, Pontificia Universidad Católica de Valparaiso, Av. Brasil 2185, Valparaíso, Chile
[17]Aix Marseille University, Université de Toulon, CNRS, IRD, MIO UM 110, Marseille, France
[18]Scientific and Technological Bioresource Nucleus, Universidad de La Frontera, Temuco, Chile
[19]Instituto de Investigaciones Agropecuarias, INIA La Platina, Casilla 439, Correo 3, Santiago, Chile





[20]Departamento de Ciencias Agropecuarias y Acuícolas  Núcleo de Investigación en Producción Alimentaria, Universidad Católica de Temuco, Casilla 15-D, Temuco, Chile

[21]Soil Resources, ETH Zurich, Zurich, Switzerland

[22]Facultad de Química y Biología, Universidad de Santiago de Chile, Av. B. O'Higgins 3363, Santiago, Chile

[23]Ministerio del Medio Ambiente, San Martín 73, Santiago, Chile

[24]Corporación Nacional Forestal (CONAF), Paseo Bulnes 285, Santiago, Chile

[25]Facultad de Ciencias Forestales y Conservación de la Naturaleza, Universidad de Chile, Santa Rosa 11315, La Pintana, Chile

[26]Instituto de Investigaciones Agropecuarias, INIA Intihuasi, Apartado Postal 36/B, La Serena, Chile

[27]Centro de Información de Recursos Naturales (CIREN), Santiago, Chile

[28]Departamento de Ciencias Ambientales y Recursos Naturales Renovables, Facultad de Ciencias Agronómicas, Universidad de Chile. Santa Rosa 11315, La Pintana, Chile

[29]Instituto de Investigaciones Agropecuarias, INIA Kampenaike, Punta Arenas, Chile

[30]ONG Suelo Sustentable, Santiago Chile

[31]Edáfica, www.edafica.cl, Santiago, Chile

[32]Oficina Regional de la FAO para América Latina y el Caribe, Bogotá, Colombia

[33]Instituto de Investigaciones Agropecuarias, INIA La Cruz, Chorrillos 86, La Cruz, Chile

[34]Instituto de Investigaciones Agropecuarias, INIA Quilamapu, Av. Vicente Méndez 515, Chillán, Chile

[35]Departamento de Ciencias Acuícolas y Agropecuarias, Universidad de Magallanes, Av. Bulnes 01855, Punta Arenas, Chile

[36]Oficina de Estudios y Políticas Agrarias (ODEPA), Ministerio de Agricultura, Teatinos 40, Santiago, Chile

[37]United Nations Convention to Combat Desertification, Regional Coordination Unit for Latin-America and the Caribbean, CELADE, Dag Hammarskjöld 3477, Vitacura, Chile

[38]Instituto de Ingeniería Agraria y Suelos, Facultad de Ciencias Agrarias, Universidad Austral, Valdivia, Chile

[39]Instituto de Ecología y Biodiversidad, Av. Libertador Bernardo O'Higgins 340, Santiago, Chile

**Correspondence:** Marco Pfeiffer (mpfeiffer@uchile.cl)

**Abstract.** One of the critical aspects in modelling soil organic carbon (SOC) predictions is the lack of access to soil information which is usually concentrated in regions of high agricultural interest. In Chile, most soil and SOC data to date is highly concentrated in 25% of the territory that has intensive agricultural or forestry use. Vast areas beyond those forms of land use have few or no soil data available. Here, we present a new database of SOC for the country, which is the result of an unprecedented national effort under the frame of the Global Soil Partnership that help to build the largest database on SOC to date in Chile named "CHLSOC" comprising 13,612 data points. This dataset is the product of the compilation from numerous sources including unpublished and difficult to access data, allowing to fill numerous spatial gaps where no SOC estimates were publicly available before. The values of SOC compiled in CHLSOC range from $6 \times 10^{-5}$ to 83.3 percent, reflecting the variety of ecosystems that exists in Chile. Profiting from the richness of geochemical, topographic and climatic variability in Chile, the dataset has the potential to inform and test models trying to predict SOC stocks and dynamics at larger spatial scales. Dataset available at https://www.doi.org/10.17605/OSF.IO/NMYS3 (Pfeiffer et al., 2019b).

# 1 Introduction

Soil organic carbon (SOC) stocks play a crucial role in the global C cycle, and equal nearly two thirds of the total terrestrial carbon stocks (Eswaran, 2000; Sarmiento and Gruber, 2002). Therefore, the contents and dynamics of the SOC stock is pool is



are essential to estimate trends in the evolution of atmospheric $CO_2$ content s to be used as an input and applied in models of global climate change (Jones et al., 2005; Davidson and Janssens, 2006). However, predictions of the SOC stock vary widely due to limited availability of soil data for remote regions and existing soil datasets being biased towards highly managed forest and agroecosystems (Duarte-Guardia et al., 2018). Chile is not exempt of those difficulties, having publicly available soil and

SOC data focused in the intensively cultivated areas of the central regions (Padarian et al., 2012, 2017). Vast areas of the country, however, are situated in the high Andean mountains, the hyperarid Atacama Desert or the inaccessible Magellanic moorlands in the Patagonian fjords for which very few soil data is available. These areas are of particular interest for SOC dynamics and stock predictions as they represent the extreme ends of a huge latitudinal climate gradient from Earth's driest extreme in the north (Atacama Desert) to the very humid conditions of Patagonian pacific margin, all flanked by the second

highest mountain range in the world (Garreaud et al., 2009; Ewing et al., 2008; Loisel and Yu, 2013).

The access to spatial explicit, consistent and reliable soil data is essential to model and map the status of soil resources globally at increasing detailed resolution in order to respond and assess world global issues (Arrouays et al., 2014; FAO, 2015; Hengl et al., 2014; Omuto et al., 2013). Furthermore, soil datasets are also one of the most important inputs for Earth System Models (ESM), to address, for example, the importance of terrestrials sinks and sources for greenhouse gases (Dai et al., 2018;

Luo et al., 2016). At the same time, soils in ESM are one of the largest sources of uncertainty (Dai et al., 2018). This is why in recent years there has being a growing effort to improve access and quality of soil datasets, being one of the key goals of pillar 4 of the global soil partnership sponsored by the Food and Agriculture Organization of the United Nations (Batjes et al., 2017; Omuto et al., 2013). In this sense, efforts to increase access to harmonized soil products, containing comparable and consistent datasets including soil carbon are highly valuable and appreciated by an increasing number of users (Arora et al., 2013; Baritz

et al., 2014; Batjes et al., 2017; Hendriks et al., 2016; Jones and Thornton, 2015; Luo et al., 2016; Maire et al., 2015).

In an unprecedented national effort, between May 2018 and April 2019 a group of professionals from 39 public and private institutions joint together to build the largest to date Chilean SOC database (CHLSOC). We built this dataset based on the compilation of different sources corresponding to soil surveys, publications, private reports, unpublished research data and cryptic documents unknown to the public or difficult to access. All data in CHLSOC (13,612 data points from 25 sources;

Table 1) is publicly available and can be downloaded for free at https://www.doi.org/10.17605/OSF.IO/NMYS3 (Pfeiffer et al., 2019b). This joint effort resulted in a consistent soil dataset of Chile to be available for the international community for analysis, exchange and interpretation.

## 2 Soil Data harmonization

### 2.1 Database sources

To build CHLSOC (Chilean Soil Organic Carbon database), we gathered, curated and harmonized 889 soil profiles and 12,723 topsoil samples from all over Chile (Table 2). Eighty nine percent of this information was unpublished and not available to the national and global scientific community. This soil information cover all the administrative regions and 16 out of 17 ecological zones of Chile (Fig. 1, Table 3). The data compiled from literature is properly referenced in Table 1. Sources





include legacy soil surveys, environmental assessment reports, research papers, private reports, theses and unpublished data facilitated by researchers. Minimum requirements for studies to enter the database were to have geographic coordinates, horizon depths and soil organic carbon content (or organic matter). If available, other soil variables such as bulk density, texture/coarse fragments, sampling depth, sampling year and measurement methods, were also included. About 20% of the total horizon

samples included information on bulk density (BLD). Where available, this information was measured using the clod method or the core (cylinder) method. Only 382 horizons (2.8%) included information about coarse fragments (CRF).

The resulting database Table 1 includes datasets of variable size, source and composition. All unpublished data sources are referenced in the database to the coauthor and group who provided the data. Examples of unpublished data sources are shown in Table 1 and include those of ODEPA with 782 points provided by J. Ramirez, METHANOBASE (Table 1), corresponding

to surface samples (0 - 25 cm) from the Magallanes Region collected during 2016 and provided by L. Cabrol and M. Barret. (Table 3). Further, 51 data points from the environmental assessment database (SEIA) were including focusing on areas with low representation, as the Andean Cordillera and the Atacama Desert. The largest contributor to CHLSOC in terms of number of data points (9,935) as well as geographical extension were the SOC dataset of the Agricultural and Livestock Service (SAG by its Spanish acronym). This dataset only includes SOC, obtained from the first 20 cm of soil by auger or excavation methods

taken by beneficiaries (farmers) of the SAG subsidy program. Another relevant data contribution was taken from soil legacy survey data compiled by CIREN (Centro de Información de Recursos Naturales) and reported as regional soil surveys, based on soil surveys that were carried from the 1960s up to 2007. The database compiled by CIREN is distributed over 177,500 km$^2$, equal to about a 24.5 % of the total Chilean territory. In total, CIREN compiled 37 soil surveys, including 540 data points, much of which are already compilations of former studies originally not referenced by CIREN (CIREN, 1996a, b, 1997a, b,

1999, 2002, 2003, 2005a, b, 2007).

## 2.2  Data harmonization processing and caveats

The assembled data has been sampled over several decades and it was no longer possible to find and verify the original data source for some of the data points. Additional uncertainty is introduced as SOC content for most samples (97%) was analyzed using wet oxidation. Only a minor fraction of the SOC data were analyzed by total combustion (CN elemental

analyzer). The difference in the analytical methods could be a source of uncertainty for future modelling initiatives which is not properly addressed in Chile on a national level. Discrepancies in the results of SOC analyses between combustion methods have identified wet combustion as an unreliable assessment method for SOC (Kumar et al., 2019). This is not yet properly addressed in Chile, as the recommended methods for SOC determination are still wet oxidation and loss on ignition, not mentioning dry combustion as a more accurate alternative (Sadzawka et al., 2006). Future initiatives of data collection should

stress consistent analytical procedures while a revision of local standards is urgent. A possible source of bias in SAG's data is that samples are taken by farmers following guidelines provided by SAG, taking a composite sampling for each parcel.



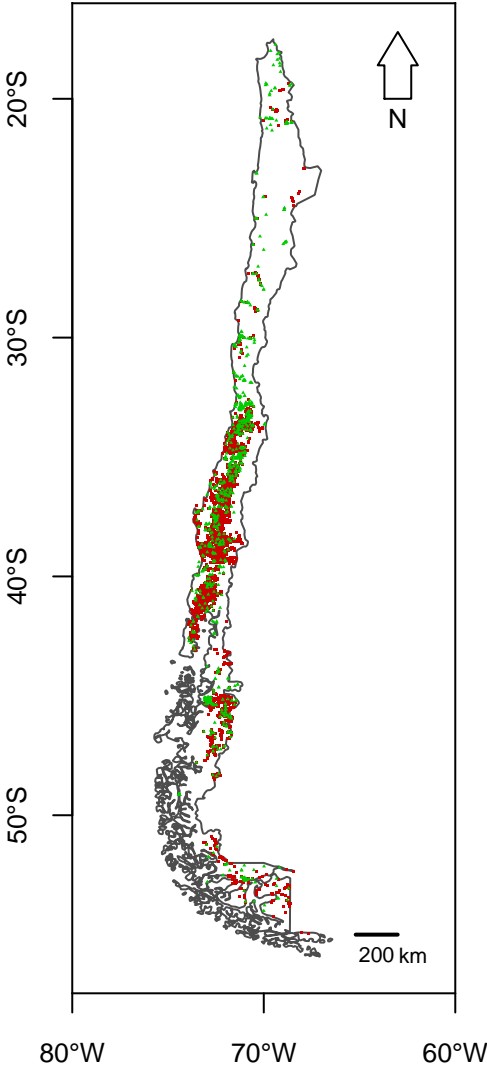

**Figure 1.** Spatial distribution of soil data points compiled in this work. Green triangles are soil profiles and red squares are topsoil samples (up to 30 cm).

## 3 Spatio-temporal distribution of the Chilean SOC dataset

### 3.1 Spatial distribution

CHLSOC currently is comprised of 13,612 points, which is a great improvement compared to former databases used in Chile for SOC assessments. For example, national SOC mapping studies (Padarian et al., 2017; Reyes Rojas et al., 2018) are based

5 almost exclusively on data of CIREN (540 points). To date, CHLSOC is the most complete data compilation for mainland Chile and can be used to reflect the influence of soil, vegetation and climatic conditions on SOC concentrations. Table 3 shows



the amount of points compiled in this work, by vegetation formation. It is important to notice that this scheme corresponds to the potential vegetation belts that originally occupied the territory and not necessarily reflects the current land use Luebert and Pliscoff (2006). In this sense, we will refer to vegetation formations as "ecosystems" as this is a more common term and avoid further specific disciplinary discussion, which is out of the reach of this work. Representativeness of each ecosystem (vegetation

formation) in CHLSOC database is based on the number of data points divided by the total coverage of the ecosystem in Chile.

About two-thirds (85.73%) of our total data are concentrated in 25% of the total country area and located in the following four ecosystems: deciduous forest, broad-leaved forest, sclerophyllous forest and thorny forest; the first two are located in the northern section of the temperate macro bioclimate, and the second two in the southern section of the mediterranean macro bioclimate zone (Moreira-Muñoz, 2011). These ecosystems are characterized by its combination of benign climate, high

quality soils and water available for irrigation, which resulted in a long history of agricultural activity and human settlements (Armesto et al., 2010). Because of this, they comprise the area that historically experienced the highest land-use conversion for agriculture, forestry and urban use in the country (Echeverría et al., 2006; Schulz et al., 2010; Arroyo et al., 2008). Deciduous forests — which comprise 14.7% of the country — is the most represented, with 52.14% of the data points collected in CHLSOC located between latitudes 35°S to 41°S.

The second biggest pool of data points, with 8.6% of the total data compiled in this work, covers evergreen forest, steppe and grassland (Table 3). These two ecosystems, that comprise 10.3% of the country area, are located between 41°S and 53°S in the Temperate macrobioclimate (Moreira-Muñoz, 2011), a thermally homogeneous territory with a considerable precipitation gradient that can reach several meters of mean annual precipitation in its western section, along the pacific coast (Garreaud et al., 2009). It contains big sections of pristine forest, with only 8% of the land being converted to other land use (Pliscoff

and Fuentes-Castillo, 2011). Most of the data collected here correspond to the eastern section of the administrative region of Aysén in Patagonia. The relatively high representation of these ecosystems in the database can be attributed to (i) the intense agricultural use of the northern section of the evergreen forest, and (ii) an unprecedented effort in soil sampling in the Aysén Region (43.5°S – 49°S) by SAG and the Agricultural Resarch Institute (INIA by its Spanish acronym) (Table 1)(Hepp and Stolpe, 2014).

The most relevant ecosystem in terms of SOC stocks for Chile correspond to the moorlands, which comprise a large area located in the Pacific coast of Patagonia where the landscape is fragmented into fjords and little islands between 44°S and 55°S . This ecosystem, which covers a significant section (9.1%) of the country area, is likely the biggest soil carbon reservoir of Chile, with an almost continuous carpet of thick peat bogs that can reach up to 5 m in some sections (Loisel and Yu, 2013; Minasny et al., 2019). Despite its relevance, most of our knowledge on soils from this ecosystem comes from its northern and

eastern borders where there is some accessibility, while soils of peatland types in the remote areas of the western fjords have almost no information available to date (20 observations in this database).

The ecosystems that comprise the Atacama Desert section of Chile (Table 3; desert, low desert scrub and desertic scrub) comprise 2.18% of the database presented in CHLSOC but correspond to 6% of the country area. Despite this low percentage, the number of data points compiled for this region (298) still constitute a great improvement compared to previous works, as the

only national work on SOC that includes data points for the Atacama Desert considered only 3 points (Padarian et al., 2017).





The scarce SOC information of this region to date is mainly because of the extreme aridity of the region and low biological activity and low amount of SOC accumulation (McKay et al., 2003). Vegetation is restricted to a narrow belt along the coast that receive water from fog, to the deep valleys that cross the desert and toward the western flank of the Andes (Moreira-Muñoz, 2011).

High altitude and mountainous regions comprise 102 data points (0.74% of the database) representing 16.2% of the country area. Two characteristic alpine vegetation formations exist in the Andean Cordillera of Chile between 18°S and 38°S, herbaceous alpine vegetation and alpine dwarf scrub. Most of the data is concentrated on the lower part (alpine dwarf scrub), while virtually no soil data is available for the higher section of the Andes (above 3000 m a.s.l.). The scarcity of soil data in this region poses a high uncertainty to assess the impact of climate change on soil C stocks as large quantities of SOC are stored is

type of ecosystem (Bockheim and Munroe, 2014).

Other vegetation formations that have few data correspond to the coniferous forest, deciduous shrubland, thorny shrubland and arborescent shrubland (Table 3). Those formations are located in areas of low forestry or agricultural interest together with a small surface area (less than 2.5% of the country).

In summary, our data compilation shows that there is an unbalance between areas of agricultural and forestry interest and

those areas beyond that land use. Three areas of high value in terms of their surface and significance, not only nationwide but also worldwide, are underrepresented in terms of soil data: High Andean Cordillera, Atacama Desert and Western Patagonia. These areas are of high interest for ecological, scientific and ecosystem service purposes. Government efforts oriented to develop soil surveys in these regions are urgent and should be promoted. In particular, a SOC inventory of western Patagonia is essential to properly assess the national stock on SOC, and the potential to include this area in carbon offsets programs.

**3.2    Temporal distribution**

In CHLSOC we provide the date of collection were this was available, so users can constrain the use of the data to a certain time frame, or only to data were the time of sampling is available. More than 90% of the included data (12,318 data points) reported the date of soil sampling. Most of the data was taken between 2006 and 2018 (Fig. 2). The high number of data from the last decade reflects that an important proportion of this database consist of recent data that can be used in doing estimations

of modern carbon contents in Chilean soils. Most of the data with years of sampling are concentrated in a short timeframe and mainly corresponds to the SAG database and to sampling efforts related to research projects such as ODEPA in 2006; INIA and NAMA in 2015; NAMA (2016), Methanobase in 2016, and a compilation of data from INIA La Platina that was compiled recently by Corradini et al. (2019). Data from CIREN (Table 1) did not report sampling date. However, as it consist of compilations from known former soil surveys, we can limit the period in which samples were collected and analyzed to

the period between 1970 and 2007. The oldest data points correspond to those collected by Holdgate (1961) in the Western Patagonian fjords in 1959.

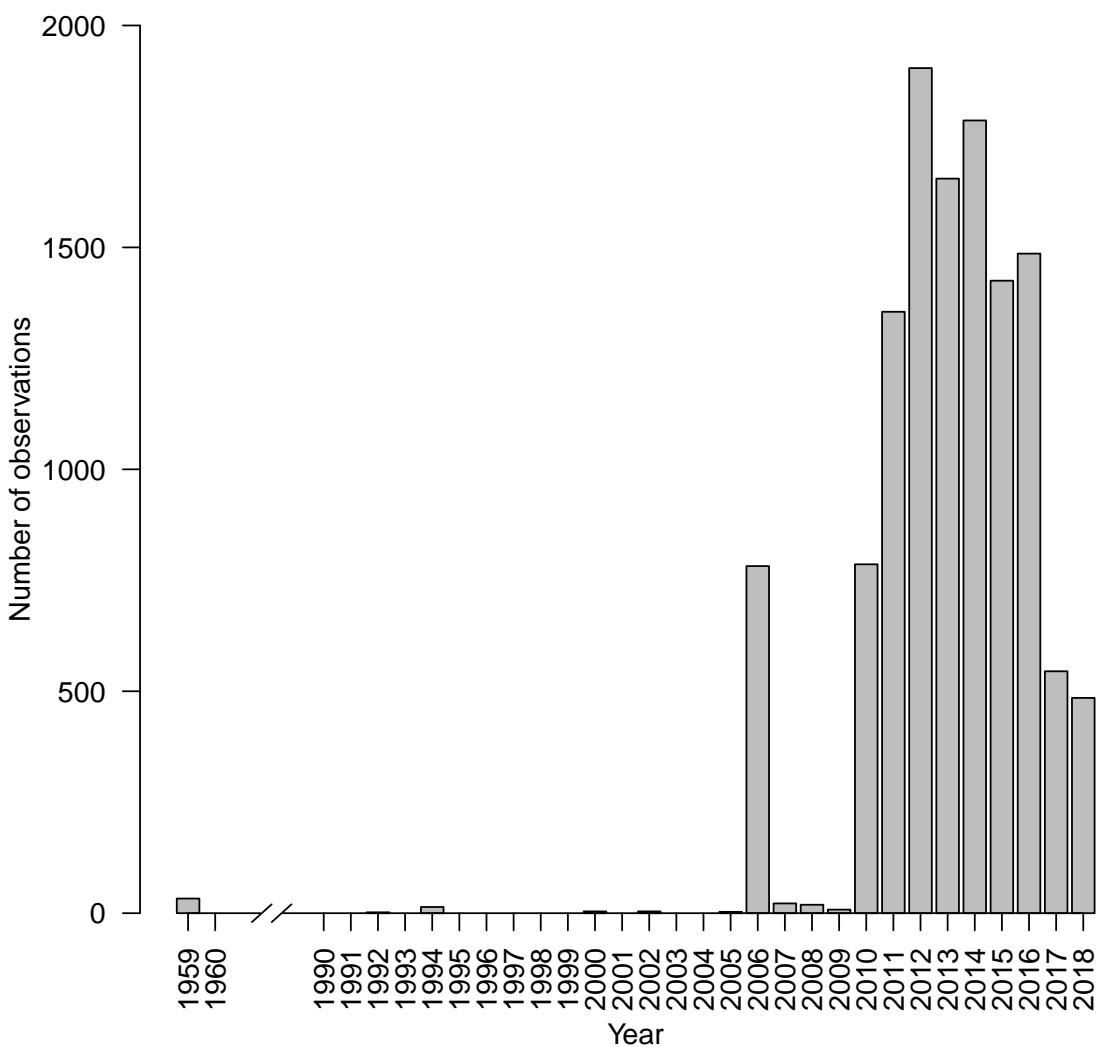

**Figure 2.** Temporal distribution of the samples included in the CHLSOC

## 4   Conclusions

The process of generating this database was a distributed data collection effort, which is a step forward under the efforts of
the GlobalSoilMap project and the guidelines of the FAO Global Soil Partnership. The database presented here increases the
public availability of SOC data of Chile ten-fold thanks to a joint effort of dozens of researchers and institutions. 89% of this



database (12,125 data points) consist of unpublished data that are now made available. CHLSOC now contains a valuable SOC representation of a mosaic of ecosystems in Chile which represents one of Earth's most extreme climate gradients. However, there are still big differences in the amount of data obtained from managed (agro)ecosystems and natural systems in areas of low population density. We like to stress the urgency to generate a discussion at a national level regarding the need of a

comprehensive soil survey program to increase the sampling in these underrepresented areas. Moreover, for a better inclusion of more data in the next versions of CHLSOC, future official CIREN soil surveys of Chile and other data sets should be encouraged to report holistic metadata covering sampling designs, locations, sampling dates and analyses methods.

*Data availability.* Data is available at https://www.doi.org/10.17605/OSF.IO/NMYS3 (Pfeiffer et al., 2019b), the data is represented by a code defining the soil name, soils from the CIREN data source are identified by a 3 letter code corresponding to the soil series and data from

other sources are identified by the site or author name. Geographical coordinates are in UTM WGS 84.

*Author contributions.* M.P., G.F.O., M.G., R.O., N.B., J.B., M.F., G.G., A.G., J.M., J.R., C.R., I.S. and S.B. designed the framework to produce the database. The manuscript was written by M.P. and J.P. with contributions from all other authors that reviewed and provided input on the manuscript.

.

*Competing interests.* The authors declare that they have no conflict of interest

*Acknowledgements.*

    Authors of unpublished data are acknowledge funding from the following sources: F.A. to FONDECYT Iniciación 11160372 and Convenio CONAF-UDeC 2015 Perturbaciones Araucaria. M. Barret and L. Cabrol to ERANet-LAC joint program ELAC2014_DCC-0092. E. Zagal and C. Muñoz to Proyecto Fondecyt 1161492. F. Olmedo and M. Guevara were supported by the Global Soil

Partnership and the South America Soil Partnership both sponsored by the Food and Agriculture Organization of the United Nations (FAO).



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



**Table 1.** Database sources used in this compilation

| Source | Samples | SOC method | BLD method | References |
|---|---|---|---|---|
| Biester | 3 | DC | Core | Biester et al. (2003) |
| CIREN | 540 | WO | Clod | CIREN (1996a); CIREN (1996b); CIREN (1997a); CIREN (1997b); CIREN (1999); CIREN (2002); CIREN (2003); CIREN (2005a); CIREN (2005b); CIREN (2007) |
| Doetterl | 22 | DC | Core | Doetterl et al. (2015) |
| EarthShape | 16 | DC | Core | Bernhard et al. (2018) |
| Filipova | 46 | WO | Core | Filipová et al. (2010) |
| Holdgate | 33 | DC | NA | Holdgate, M. 1961 |
| INIA | 1663 | WO | Clod; NA | Hepp and Stolpe (2014); Besoain et al. (2000);Corradini et al. (2019, 2017); Corradini, F. Unpublished data; Ovalle, C., INIA, Unpublished data; Panichini et al. (2012, 2017); Pfeiffer, M., Ivelic-Sáez, J., Valle, S., Unpublished data |
| McCulloch | 2 | WO | Clod | McCulloch and Davies (2001) |
| Methanobase | 37 | DC | NA | Cabrol, L. & Barret, M., Unpublished data |
| Mörchen | 12 | DC | Excavation | Mörchen et al. (2019) |
| ODEPA | 782 | WO | NA | Ramirez, J., Unpublished data. |
| PUC | 24 | WO | Clod; Core | Arellano, E., unpublisehd data |
| Quade | 16 | DC | NA | Quade et al. (2007) |
| SAG | 9935 | WO | NA | Gomez, A., Unpublished data; Osorio, R., Bustamante, N., Unpublished Data |
| Schuller | 14 | WO | Core | Schuller et al. (2004) |
| SEIA | 51 | WO | Clod | Riveras, N., unpublished data |
| UACH | 3 | WO | Clod; Core | Gerding and Thiers (2002); Pfeiffer, M., Ivelic-Sáez, J., Valle, S., Unpublished data |
| UAP | 85 | WO | Clod | Briceño M. unpublished data; Delatorre et al. (2008); Ehleringer et al. (1992) |
| UCB | 8 | DC; WO | Clod | Ewing et al. (2006, 2008); Finstad et al. (2018); Pfeiffer et al. (2019a); Pfeiffer, M. Unpublished data; Pfeiffer, M., Ivelic-Sáez, J., Valle, S., Unpublished data |
| UChile | 198 | WO | Clod; Core; NA | Norambuena (2000); Pfeiffer et al. (2012); Casanova, M. Unpublished data; Casanova, M., Salazar, O. Unpublished data;Fuentes et al. (2014); Fuentes, JP, unpublished data; Galleguillos, M., unpublished data;Kirberg (2014); Martínez et al. (2017); Mashalaba, L., unpublished data;Perez, J., Galleguillos, M., Unpublished data; Seguel et al. (2015); Seguel, O. Unpublished data;Soto et al. (2015) |
| UCT | 5 | WO | Core | Curaqueo et al. (2010, 2011, 2014); Curaqueo. Unpublished data |
| UDEC | 86 | DC; WO | Core; NA | Hepp and Stolpe (2014); Aburto, F. (Unpublished data); Zagal, E., Munoz, C., Doeterl, Unpublished data |
| UFRO | 16 | WO | NA | Garrido and Matus (2012) |
| UMAG | 11 | DC | Core | Radic et al. (2013) |
| Ziolkowski | 4 | DC | NA | Ziolkowski et al. (2013) |

BLD: bulk density; WO: wet oxidation; DC: dry combustion; NA: data not provided



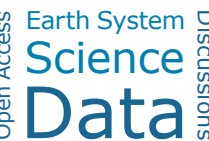

**Table 2.** Summary of the soil points included in the Chilean Soil Organic Carbon Dataset (CHLSOC)

| Variable | value |
|---|---|
| Number of profiles | 889 |
| Number of topsoil samples | 12,723 |
| SOC measurements | 16,884 |
| SOC using method wet oxidation | 16,363 |
| SOC using method dry combustion | 521 |
| Minimum SOC (%) | $6\times10^{-5}$ |
| Maximum SOC (%) | 83.30 |
| BLD measurements | 2,757 |
| BLD using core method | 533 |
| BLD using clod method | 2,224 |
| Minimum BLD (g/cm$^3$) | 0.03 |
| Maximum BLD (g/cm$^3$) | 2.38 |
| CRF measurements | 382 |
| Minimum CRF (%) | 0.00 |
| Maximum CRF (%) | 78.31 |

SOC: soil organic carbon; BLD: bulk density; CRF: coarse
fragments; topsoil considers points with surface samples only
(<30 cm)



**Table 3.** Distribution of SOC data points per ecosystem (vegetation formation) according to Luebert and Pliscoff (2006)

| Vegetation formation | Data points | Country area (%) | Representativeness index (points per % area) | SOC (mean) | SOC (min) | SOC (max) | SOC (sd) |
|---|---|---|---|---|---|---|---|
| Deciduous forest | 7098 | 14.70 | 482.86 | 8.01 | 0.00 | 83.30 | 3.91 |
| Sclerophyllus forest | 2544 | 5.20 | 489.23 | 2.66 | 0.00 | 15.61 | 1.87 |
| Thorny forest | 1392 | 2.80 | 497.14 | 1.90 | 0.00 | 20.70 | 1.50 |
| Broad-leaved forest | 645 | 1.90 | 339.47 | 12.02 | 0.15 | 25.75 | 5.11 |
| Coniferous forest | 94 | 2.30 | 40.87 | 6.02 | 0.10 | 25.00 | 3.40 |
| Evergreen forest | 828 | 6.90 | 120.00 | 11.70 | 0.01 | 81.19 | 7.89 |
| Desert | 47 | 7.70 | 6.10 | 1.63 | 0.00 | 15.00 | 1.84 |
| Steppe and grassland | 343 | 3.40 | 100.88 | 6.02 | 0.00 | 56.70 | 10.87 |
| Herbaceous alpine vegetation | 2 | 2.40 | 0.83 | 5.11 | 5.01 | 5.20 | 0.13 |
| Evergreen shrubland | 0 | 0.30 | 0.00 | – | – | – | – |
| Alpine dwarf scrub | 100 | 13.80 | 7.25 | 1.42 | 0.01 | 41.60 | 6.09 |
| Low desert scrub | 20 | 8.70 | 2.30 | 0.92 | 0.02 | 25.40 | 2.94 |
| Deciduous shrubland | 2 | 2.30 | 0.87 | 4.47 | 2.20 | 10.19 | 3.19 |
| Desertic scrub | 231 | 9.50 | 24.32 | 1.19 | 0.00 | 22.60 | 1.69 |
| Thorny shrubland | 17 | 0.30 | 56.67 | 1.11 | 0.20 | 2.49 | 0.61 |
| Arborescent shrubland | 158 | 1.00 | 158.00 | 5.26 | 0.08 | 42.50 | 5.70 |
| Moorland | 20 | 9.10 | 2.20 | 43.86 | 1.74 | 57.71 | 21.10 |

Percentages of surface area and English names for vegetation formations were taken from Pliscoff and Fuentes-Castillo (2011)