# Peer review of "CHLSOC: The Chilean Soil Organic Carbon database, a multi-institutional collaborative effort"

_Earth System Science Data, 2019_

## Referee Comment (RC1) · Sergey Zimov (Referee) · 30 Oct 2019

The manuscript opens access to a huge amount of fresh data of SOC gathered from longest meridian transect from tropic to tundra. I think it should be published as soon as possible and waste no many words and time for it's edition and notes. ESSD is a proper place for the manuscript.

---

## Referee Comment (RC2) · Anonymous Referee #2 · 31 Oct 2019

Comments by anonymous referee on: CHLSOC: The Chilean Soil Organic Carbon database, a multi-institutional collaborative effort

Data:

1. I would prefer unabbreviated column headings with units or an explanation table for column headings and units in the data paper.
2. There are 0 values in columns oc and crf in the data – is this possible?

Discussion paper:

[revised manuscript text omitted]

**Commented [CJ4]:** Suggest rewriting this. Doesn't quite make sense.

---

## Author Comment (AC1) · 18 Nov 2019

We would like to sincerely thank Dr. Sergey Zimov for his positive assessment of this work.

———————————————————

---

## Referee Comment (RC3) · Anonymous Referee #2 · 19 Nov 2019

Thank you for taking on board my comments and suggestions and updating the manuscript. I think the changes you have made mean that the paper flows nicely. It is a good explanation of the associated data and following a thorough read, I now feel I understand the work that you have done. I have no further suggested changes.

---

## Short Comment (SC1) · 19 Nov 2019

I appreciate your feedback and prompt reply.

Best,

Marco

---

## Author Comment (AC2) · 19 Nov 2019

We appreciate the positive evaluation of referee #2 of our manuscript, as well as the time taken in copy editing the manuscript.

1. I would prefer unabbreviated column headings with units or an explanation table for column headings and units in the data paper.

Response: We Include a table (Appendix 1) with a short description plus units used in the database

2. There are 0 values in columns oc and crf in the data – is this possible?

Response: Those values are reported as 0 by the original published documents which

are referenced in the database. We try to give as much information as possible as well as inform possible biases to the users so they can use the database according to their own criteria.

In addition we proof read the entire article (see attached document) and made changes following the referee advice plus additional changes in the writing to improve the flow of the text.

Please also note the supplement to this comment:
https://www.earth-syst-sci-data-discuss.net/essd-2019-161/essd-2019-161-AC2-supplement.pdf

———————————————————

---

## Referee Comment (RC4) · Anonymous Referee #3 · 24 Nov 2019

I have read through the manuscript "CHLSOC: The Chilean Soil Organic Carbon database, a multi-institutional collaborative effort" submitted to ESSD. The CHISOC dataset compiled a huge amount of SOC data, which is an important data resource. I think the database could be published in your Journal.

I am interested to see how consistency between this data and some global soil organic carbon datasets, for example: The HWSD has SOC data, please see their data at: https://daac.ornl.gov/cgi-bin/dsviewer.pl?ds_id=1247 The updated global carbon map: https://esdac.jrc.ec.europa.eu/ESDB_Archive/octop/Resources/Global_OC_Poster.pdf It should be easy to link your dataset (by latitude and longitude) with above two datasets, and analyze the consistency between them.

This will be a great contribution to evaluate the data quality of global SOC datasets;

[Figure]

[Figure]

meanwhile, it is also a good way to evaluate the quality of CHISOC.

Specific comments Abstract Line 2: do you mean both "soil and SOC data" are highly concentrated in 25% of the territory, or do you mean "soil SOC data"? In my opinion, it makes more sense to say "soil SOC data". Please check. Line 7: "difficult to access data" sounds not the best expression, change to "inaccessible data"? But I am not a native English speaker, please check with the native speaker.

Introduction Line 2: "the contents and dynamics of the SOC stock is pool is are essential to..." please check this sentence. Line 3: "atmospheric $CO_2$ content s to be used as an input", there is a space in the word "content s", please delete it.

Figure Figure 2. I can understand why before 2005, there are not much data. But why 2007, 2008, and 2009 do not have many data?

---

## Referee Comment (RC5) · Anonymous Referee #4 · 24 Nov 2019

*There are still areas where the grammar and sentence structure needs work, especially in the first few paragraphs.

*page 3, I don't think the most of the paragraph that begins at line 65, where the number of data points contributed by various people is mentioned, adds much to the manuscript. I'd suggest shortening this paragraph just to the first two sentences and general information that the data came from a variety of sources, including areas of low representation (i.e METHANOBASE and SEIA data) and by scientists as well as beneficiaries (farmers) of the Agricultural and Livestock Service (SAG by its Spanish acronym) subsidy program.

*page 3, line 94: It would be better to introduce this section with a topic sentence saying that there are several caveats users should be aware of with these data and

then list these caveats in sentences starting with first, second, and finally, so the reader understands when they authors are transitioning from one idea to another

*page 3, line 97: what level of uncertainty is added by using the wet oxidation method? Why? Is there a reference you could add here where users of the data could learn more about this issue if they wanted to? Oh, some of that information is at line 103. These sentences need to be together. And I'd like more information explicitly given to the reader about the potential errors introduced by wet oxidation (too high, too low?) so they don't need to go to that reference to figure that information out.

*page 3, line 102: I don't think you need include "which is not properly addressed in Chile on a national level", especially since the authors bring this up again in page 4, line 1.

*Table 2: I'm suspicious of the SOC values based on what's listed in Table on as minimum and maximum values. First, the minimum value is listed as 0.00006 % C. I don't know any method that can accurately measure C levels that low. The maximum value is also listed as over 80 % C. I work in highly organic C soils and I have never seen a % C value higher than about 60 % and that was in a burned area. In addition, based on Table 3, these point are found in areas that are not known for high C soils. Should these data have been excluded during a QC process. Did you do any QCing? Or did you accept all data given to you? Either way, it should be explicitly stated that the data were or were not reviewed (and if they were how they were evaluated should also be included).

*I notice where the data are online there is no metadata file. I would suggest you add a meta data file to the online location of the data so that users who come upon the data without finding this reference are able to use it. (This is not something that needs to happen before the paper is published, but a recommendation for future users.)

---

## Author Comment (AC3) · 20 Dec 2019

To Referee #3

We appreciate the comments and suggestions to our work, as well as the positive evaluation of our manuscript. Please consider that the English language in this version of the manuscript was proof read by a professional proofreader.

Regarding the specific comments, please find bellow our detailed response to comments.

Reviewer comment: I am interested to see how consistency between this data and some global soil organic carbon datasets, for example: The HWSD has SOC data, please see their data at:

[Figure]

https://daac.ornl.gov/cgi-bin/dsviewer.pl?ds_id=1247 The updated global carbon map: https://esdac.jrc.ec.europa.eu/ESDB_Archive/octop/Resources/Global_OC_Poster.pdf It should be easy to link your dataset (by latitude and longitude) with above two datasets, and analyze the consistency between them. This will be a great contribution to evaluate the data quality of global SOC datasets; meanwhile, it is also a good way to evaluate the quality of CHISOC.

Authors' Response: As suggested, we checked the databases indicated. Both maps, used a very small number of point data from Chile, which happens with most of available global maps. For instance, WoSIS Soil Profile Database has only 45 data points for Chile (Batjes et al., 2019), which are the same used for both global maps indicated by the reviewer. We would like to highlight that this is the first time Chile generates and publishes a consistent soil organic carbon database. As such, we consider that a comparison with other datasets is out of the scope of this paper, especially considering that the datasets mentioned are gridded maps, generated by modelling or interpolating fields measurements. A comparison will only assess the quality of those maps and not the quality of CHISOC, which is a soil profile collection.

We consider that any data to date (databases or maps generated from them) are of inferior quality since they do not have enough samples to represent the pedodiversity of Chile. After a quick analysis, it is possible to show that a map such as HWSD does not represent accurately a local scale (See Attached Figure).

Since we consider that is not possible to compare it in terms of quality (there is not comparable database to date), we added a small comparison in terms of the number of samples (WoSIS) in the introduction as follow: "This work ended up with an harmonized dataset of 13,612 points, which is a great improvement considering that up to date harmonized data on SOC for Chile include 45 points in WOSIS (Batjes et al., 2017)." - - - - - - - - - -

Reviewer comment: Specific comments Abstract Line 2: do you mean both "soil and

SOC data" are highly concentrated in 25% of the territory, or do you mean "soil SOC data"? In my opinion, it makes more sense to say "soil SOC data".

Authors' Response: We change the phrase to "To date, in Chile, a large proportion of the soil SOC data has been collected in areas of intensive agricultural or forestry use, however, vast areas beyond these forms of land use have few or no soil data available." - - - - - - - - -

Reviewer comment: Please check. Line 7: "dificult to access data" sounds not the best expression, change to "inaccessible data"? But I am not a native English speaker, please check with the native speaker.

Authors' Response: We checked with a native speaker and reorder the phrase to make it clearer: "This dataset is the product of the compilation from numerous sources including unpublished and difficult to access data, allowing to fill numerous spatial gaps where no SOC estimates were publicly available before." - - - - - - - - -

Reviewer comment: Introduction Line 2: "the contents and dynamics of the SOC stock is pool is are essential to..." please check this sentence. Line 3: "atmospheric CO2 content s to be used as an input", there is a space in the word "content s", please delete it.

Authors' Response: We rephrase the sentence as follow "knowledge of the contents and dynamics of the SOC stock is essential for estimating trends in the evolution of atmospheric carbon dioxide ($CO_2$), to be used as an input and applied to models of global climate change". - - - - - - - - -

Reviewer comment: Figure 2. I can understand why before 2005, there are not much data. But why 2007, 2008, and 2009 do not have many data?

Authors' Response: Thanks for noting this point. This is mainly related to the databases we were able to access during this compilation effort. We added more information in section 3.2 "Temporal distribution" in order to address this point: "The date of sample

collection is provided in more than 90% of the included data (12,318 data points). The majority of points were sampled in 2006 and between 2010 and 2018 (Figure 2). The high number of data from the last decade enables users to estimate modern carbon in Chilean soils. Most of the data that report the year in which it was sampled is concentrated in a short timeframe and mainly corresponds to the SAG database (2010-2018) and to sampling efforts related to research projects such as ODEPA in 2006 and INIA (mainly 2015-2018)."
* * *
[Figure]

Comparison of random HWSD cells (0.05 degree per pixel*) with different carbon content
values with values of datapoints in CLSOC that fall into that cells.

*This corresponds to 180 arcsec  or 5km at the Ecuator

**Fig. 1.** Comparison of random HWSD cells (0.05 degree per pixel*) with different carbon content values with values of datapoints in CLSOC that fall into that cells.

---

## Author Comment (AC4) · 20 Dec 2019

To Referee #4

We appreciate the time taken to identify those points that need more work. Please consider that the English language in this version of the manuscript was proof read by a professional.

- - - - - - - - -

Reviewer comment: There are still areas where the grammar and sentence structure needs work, especially in the first few paragraphs.

Authors' Response: Then entire document was proofread by a professional. - - - - - - -

[Figure]

- -

Reviewer comment: *page 3, I don't think the most of the paragraph that begins at line 65, where the number of data points contributed by various people is mentioned, adds much to the manuscript. I'd suggest shortening this paragraph just to the first two sentences and general information that the data came from a variety of sources, including areas of low representation (i.e METHANOBASE and SEIA data) and by scientists as well as beneficiaries (farmers) of the Agricultural and Livestock Service (SAG by its Spanish acronym) subsidy program.

Authors' Response: The paragraph was shortened and rewritten for clarity.

- - - - - - - - -

Reviewer comment: page 3, line 94: It would be better to introduce this section with a topic sentence saying that there are several caveats users should be aware of with these data and then list these caveats in sentences starting with first, second, and finally, so the reader understands when they authors are transitioning from one idea to another

Authors' Response: We rewrite as suggested and think this will greatly improve clarity of the paragraph. The final paragraph ended as follows: "The assembled data was sampled over several decades and compiled by different authors or institutions. We would like to mention the following warnings to the data users: first, for some data points it was not possible to find or verify the original data source. Second, a potential source of uncertainty may be the analytical method employed for analysis; for most samples (97%), SOC content was analyzed using the wet oxidation method and a small number were analyzed by total combustion (CN elemental analyzer). Discrepancies in SOC results between combustion methods have identified wet combustion as a less reliable assessment method for SOC, as it tends to underestimate organic carbon at higher SOC contents (Kumar et al., 2019), and potentially overestimate in highly reduced soils (Chatterjee et al., 2009). This issue has not been addressed in Chile to

date. The recommended methods for SOC determination are currently wet oxidation and loss on ignition, however, dry combustion is a more accurate alternative (Sadzawka et al., 2006). Future data collection initiatives should stress consistent analytical procedures as a revision of local standards is urgently required. Finally, a possible source of bias in data from SAG is the fact that samples were taken by farmers following SAG guidelines where a composite sampling is taken for each parcel."

- - - - - - - - - Reviewer comment: page 3, line 97: what level of uncertainty is added by using the wet oxidation method? Why? Is there a reference you could add here where users of the data could learn more about this issue if they wanted to? Oh, some of that information is at line 103. These sentences need to be together. And I'd like more information explicitly given to the reader about the potential errors introduced by wet oxidation (too high, too low?) so they don't need to go to that reference to figure that information out.

Authors' Response: We appreciate this suggestion and addressed the issue by adding more information and a comprehensive review on the methodology as a reference. The phrase changed as follows: "Discrepancies in SOC results between combustion methods have identified wet combustion as a less reliable assessment method for SOC, as it tends to underestimate organic carbon at higher SOC contents (Kumar et al., 2019), and potentially overestimate in highly reduced soils (Chatterjee et al., 2009)."

- - - - - - - - - Reviewer comment: page 3, line 102: I don't think you need include "which is not properly addressed in Chile on a national level", especially since the authors bring this up again in page 4, line 1.

Authors' Response: The phrase was eliminated

- - - - - - - - -

Reviewer comment: Table 2: I'm suspicious of the SOC values based on what's listed in Table on as minimum and maximum values. First, the minimum value is listed as

0.00006 % C. I don't know any method that can accurately measure C levels that low. The maximum value is also listed as over 80 % C. I work in highly organic C soils and I have never seen a % C value higher than about 60 % and that was in a burned area. In addition, based on Table 3, these point are found in areas that are not known for high C soils. Should these data have been excluded during a QC process. Did you do any QCing? Or did you accept all data given to you? Either way, it should be explicitly stated that the data were or were not reviewed (and if they were how they were evaluated should also be included).

Authors' Response: Regarding the values, we reported the values as they are in the original sources. The values mentioned by the reviewer are published values and methods can be checked by the users in the original source. Regarding the methodology used to obtain very low values as the one reported in table 3, it is worth to mention that when using AMS to determine isotopic composition of the SOC it is possible to obtain very low values, which is the case of that particular lowest value in the database measured and reported by Ewing et al., (2006, 2008). We include AMS as a dry combustion method as signaled by reviews of SOC methodology (e.g. Chatterjee et al., 2009). Other very low values exist in the database for the Atacama desert, some of them corresponding to a recent article published by Mörchen et al. (2019) and included in the database; in this study they used a Solu TOC Cube (Elementar Analysensysteme, Hanau, Germany), and extended up to 5000mg of sample weight for very low C contents. Regarding the 83.3% value reported in table 3, this corresponds to a sample obtained from a Sphagnum peat bog by one of our coauthors (J.P. Fuentes), who performed the wet oxidation method. We think this additional information is not necessary to address in the paper as it can be obtained by the users of the database directly from the sources. Anyway, if the reviewer thinks a paragraph as the above mentioned is necessary and the editor concurs, we can add this information to the manuscript.

- - - - - - - - -

Reviewer comment: I notice where the data are online there is no metadata file. I

would suggest you add a meta data file to the online location of the data so that users who come upon the data without finding this reference are able to use it. (This is not something that needs to happen before the paper is published, but a recommendation for future users.)

Authors' Response: A metadata file is being prepared to be added to the database repository.